# High-Coverage Satellite-Based Coastal Bathymetry through a Fusion of Physical and Learning Methods

**Céline Danilo ***[ID] and Farid Melgani[ID]

Department of Information Engineering and Computer Science, University of Trento, via Sommarive, 9, 38123 Trento, Italy; farid.melgani@unitn.it

*   Correspondence: celine.danilo@gmail.com

**Abstract:** An up-to-date knowledge of water depth is essential for a wide range of coastal activities, such as navigation, fishing, study of coastal erosion, or the observation of the rise of water levels due to climate change. This paper presents a coastal bathymetry estimation method that takes a single satellite acquisition as input, aimed at scenarios where in situ data are not available or would be too costly to obtain. The method uses free multispectral images that are easy to obtain for any region of the globe from sources such as the Sentinel-2 or Landsat-8 satellites. In order to address the shortcomings of existing image-only approaches (low resolution, scarce spatial coverage especially in the shallow water zones, dependence on specific physical conditions) we derive a new bathymetry estimation approach that combines a physical wave model with a statistical method based on *Gaussian Process Regression* learned in an unsupervised way. The resulting system is able to provide a nearly complete coverage of the 2–12-m-depth zone at a resolution of 80 m. Evaluated on three sites around the Hawaiian Islands, our method obtained estimates with a correlation coefficient in the range of 0.7–0.9. Furthermore, the trained models provide equally good results in nearby zones that lack exploitable waves, extending the scope of applicability of the method.

**Keywords:** coastal bathymetry; machine learning; unsupervised learning; linear wave model; Gaussian Process regression; multispectral images; Hawaiian Islands

---

## 1. Introduction

Sea coasts are extremely dynamic and fragile regions that are constantly exposed to diverse natural and anthropogenic phenomena, in particular in the context of climate change. Thus, a precise monitoring of coastal dynamics (such as erosion, waves, currents, moving seabeds, or rising sea levels), and specifically of coastal bathymetry, is crucial for activities such as urban planning, fishing, navigation, or the protection of environment.

Coastal bathymetry traditionally relies on in situ campaigns using sonar or airborne LIDAR. Being expensive and time-consuming, these methods are infrequently used and cannot be applied on a regular basis. Remote sensing techniques using satellite images propose a cost-effective alternative: one sensor covers a large geographical area with a repetitive time interval of typically 5–10 days. So far two main approaches have been put forth for the purpose of coastal bathymetry:

*   the *water colour inversion model* that relies on the extinction of light with depth;
*   the *linear wave model* that relies on a physical relationship between the water depth and the wavelength of waves approaching the coast.

The water colour inversion model has been in use since the 1970s [1–6]. This method provides coastal bathymetry with a high spatial resolution (equal to the resolution of the images), but results greatly depend on water quality, seabed reflection, and atmospheric effects. The mean difference between measured and estimated depths is around 20% between 0 and 6 m [4]. The principle is widely used to propose bathymetry products until 35 m of depth in clear waters [7]. One of its major limitations is its need for in situ measurements for calibration.

The linear wave model, in contrast, does not need in situ data for calibration and is robust with respect to water or seabed quality (turbid or clear waters, sandy or rocky seabed, etc.). So far two main types of approaches based on the linear wave model have been proposed: [8,9] require the availability of two images within a very short time interval in order to compute wave properties, while [10–13] rely on a single acquisition. While the methods of the first kind are generally able to provide good coverage and resolution, the two-image constraint limits input data sources to a small number of instruments with non-free availability (e.g., Pléiades, Worldview).

In [12] the authors have implemented a single-image solution using a *wave tracing* technique that was evaluated to have a standard deviation of less than 15% of the observed depth between the surf zone until as far as 20 m, for sites with a direct exposure to the swell and with an absence of clouds (the latter being an obvious limitation to the exploitation of optical imagery). However, this method, as well as all others based on a single image, also have important limitations in terms of *coverage* that may be dissuasive for end users:

- it depends on the presence of swell and, consequently, it cannot cover acquisitions of calm zones;
- the estimations provide an incomplete spatial coverage: the propagation of waves is often irregular and does not always reach the shore;
- the resolution is low (about 500 m at best).

In this paper, our main objective is to eliminate or reduce these limitations in order to be able to offer an exploitable product. Our key contribution is the combination of the linear wave model (that we will call *physical model* as it is based on wave physics) with a statistical model using *Gaussian Process Regression* (GPR) in an unsupervised setup. First known in the geostatistics community as *kriging*, GPR is a machine learning approach introduced by [14] that has been successfully used in recent Earth Observation applications [15,16].

Our goal is to obtain a water depth product on a regular grid with a resolution of less than 100 m that better corresponds to the needs of real-world applications. We achieve this result by the unsupervised training of a GPR model with input features extracted from multispectral bands of a single acquisition. This approach is an alternative way to exploit image reflectance and, as such, can be considered as a special case of the water color inversion model. We are thus able to combine the approach based on wave physics with the one based on water colour while maintaining independence of in situ measurements and relying on very little to no human supervision.

Section 2 of our paper gives an overview of the combined physical–statistical method. Sections 3–5 describe the three main components of the method, namely physical estimation, statistical model optimisation, and statistical estimation. Section 6 evaluates the method over acquisitions around the Hawaii Island of Oahu. In the goal of extending the scope of applicability of our method to images without exploitable waves, Section 7 evaluates and discusses the transfer of our trained statistical GPR models, both in space (to surrounding zones at the same time of acquisition) and in time (to earlier or later acquisitions of the same zone).

## 2. Method Overview

Our method combines two approaches: a physical estimation based on *linear wave theory* and a statistical estimation based on *water colour inversion*. The fundamental hypothesis is that, if input conditions are met (as detailed below), an initial application of the physical model produces an output that, while incomplete in spatial coverage, is reliable enough to train and subsequently apply a statistical regression method in an unsupervised manner. In order to ensure a sufficient accuracy of the unsupervised training data, only results with a high enough reliability are used from the output of the physical model.

Accordingly, the method is divided into three main steps, as shown in Figure 1:

1.  *physical bathymetry estimation:* the physical model is applied to the input satellite image, producing physical bathymetry estimates as output;
2.  *statistical model optimisation:* from a set of candidate statistical GPR models, an optimal one is trained using the input image as well as the physical bathymetry estimates;
3.  *statistical bathymetry estimation:* the optimal statistical model is applied to the multispectral image to obtain high-resolution, high-coverage estimates; the same model can subsequently be reapplied to obtain bathymetry estimates for new images that are spatially or temporally related (i.e., close) to the original image.

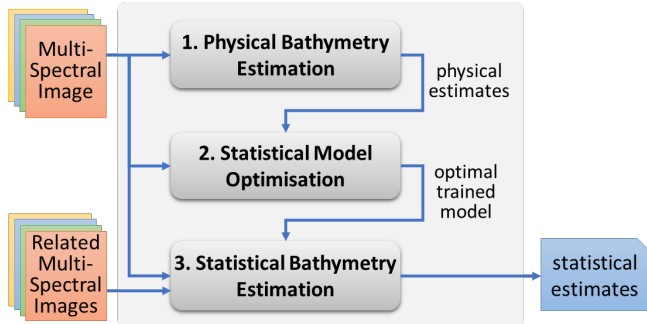

**Figure 1.** High-level diagram of the bathymetry estimation method.

The input of our method consists of a single optical acquisition. For our research we used images taken by the MSI sensor onboard Sentinel-2 (A and B), but there is no theoretical constraint on using other sources with different characteristics (For example, in [17], the authors showed that estimating bathymetry from satellite is optimal at a resolution between 10 and 20 m and at least three spectral bands (one between 485 and 550 nm, another one between 550 and 580 nm and a last one in the NIR domain)). On the 13 channels provided by Sentinel-2, we used four with a 10-m resolution: *red* (wavelength between 457.5 and 522.5 nm), *green* (542.5–577.5 nm), *blue* (650–680 nm), and *near-infrared* (784.5 to 899.5 nm) . Nevertheless, the method remains applicable if the set of channels provided by the sensor are different. We are not applying atmospheric corrections and are using the images as provided by ESA into the level-1C product.

In terms of natural conditions, a low cloud coverage is obviously necessary for the method to be applicable. Furthermore, the presence of swell with a period of at least 8 s is required by the physical model (step 1 in Figure 1), but not by the statistical one (step 3 in Figure 1).

## 3. Physical Bathymetry Estimation

Based on *linear wave theory* described in [18], this method has been explored since as early as the Second World War. The theory describes how wavelengths decrease as the waves approach the coast. Knowing the wavelength $\lambda$ and the wave period $T$, it is possible to estimate the water depth $h$:

$$h = \frac{\lambda}{2\pi} \text{atanh} \left( \frac{2\pi\lambda}{gT^2} \right) \tag{1}$$

where $g$ is the standard acceleration due to gravity.

In [12] the authors have presented an original method to estimate both the wavelength and the wave period. Using a wave tracing approach (Figure 2), the water depth can be deduced from the image itself, without relying on external information providing the wave period.

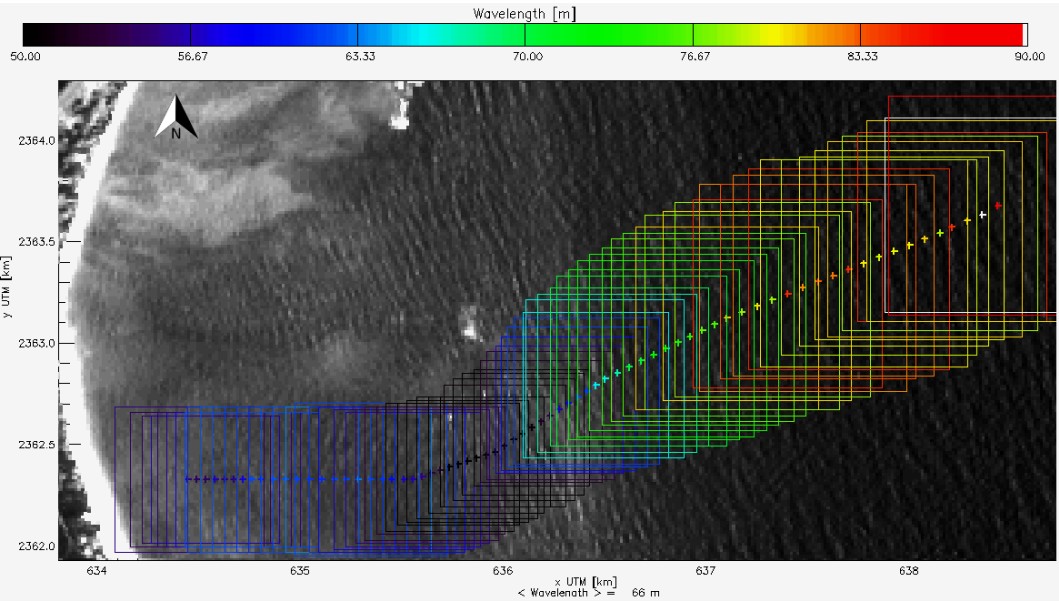

**Figure 2.** Example of the physical model results on Landsat-8 panchromatic image, 15 m resolution, acquired on the 2 June 2014 on Waimanalo Beach, Hawaii.

We apply the model only to the near-infrared band of the multispectral image. The reason is that this band is the least sensitive to changes of reflectance due to water turbidity and the type of seabed, phenomena that can potentially interfere with the wave tracing method of the physical model. As input parameters we use those developed in [12].

The model provides bathymetry estimates within the range of about 3–25 m of depth. Under 3 m, the linearity of the physical model does not hold anymore. Above 25 m, the shoaling effect may be compromised with other interactions. Furthermore, the spatial coverage of the output is irregular: firstly, resolution is proportional to the wavelength of the swell which, in turn, depends on the water depth. In practice, on Sentinel-2 images the highest reachable resolution is around 500 m. Secondly, the wave tracing method produces its output along the wave rays which converge and diverge in space following the refraction and diffraction of waves due to coastal geometry.

While suboptimal for direct use in real-world applications in terms of coverage and resolution, the physical estimates are remarkably precise. In [12], we found that, for sites with a direct exposure to the swell and with an absence of clouds, the standard deviation of bathymetry produced by the physical

model was less than 15% of the observed depth between the surf zone and where depth reaches about 25 m. The precision of these results make them suitable for further statistical processing.

For a more detailed presentation of the theory and implementation of the physical model, we refer the reader to the paper [12].

## 4. Statistical Model Optimisation

We define an unsupervised machine learning setup using Gaussian Process Regression where the training data consists of the physical bathymetry estimates and the reflectance values provided by the image at the same locations. Estimation then consists of feeding multispectral reflectance data taken from the entire image into the GPR estimator. This can be understood as a learning-based modelling of the well-known *water colour inversion method*. Contrarily to the physical model, we designed the output of the statistical model to provide results that are exploitable in practical applications, which involves a high enough resolution and a regular and complete coverage of the shallow water zone. We set the output resolution to 80 m on a regular grid as a compromise: while it is slightly lower than the pixel-level resolution of the input image (e.g., 10 m for Sentinel-2), this patch size allows us to smooth the effect of waves on reflectance and improve the overall output quality.

While the water colour inversion method conventionally needs in situ calibration to work, i.e., gathering of ground truth and ancillary data such as water turbidity or weather conditions, our approach uses the output of the physical model in lieu of ground truth and considers ancillary data as latent variables of the machine learning model. This lack of proper ground truth data, however, needs to be compensated by a careful optimisation of the training parameters and input, which is the subject of the current section.

The optimisation phase consists of four substeps, as shown in Figure 3:

1. co-location of reflectance values with the physical bathymetry estimates;
2. generation of several *candidate training sets* used as input for the subsequent training of candidate models;
3. training of candidate Gaussian Process Regression models on each candidate training set;
4. computation of statistical GPR estimates produced by each candidate model, followed by the selection of the best-performing model.

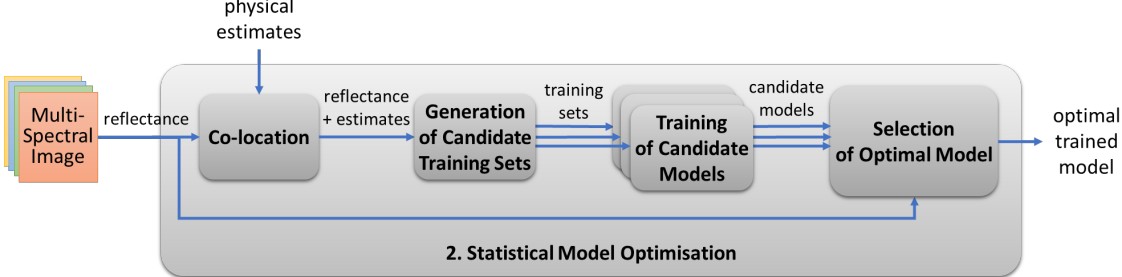

**Figure 3.** Substeps of statistical model optimisation.

All of these steps are run in a fully automated manner, allowing the computation of an optimised estimator without any human intervention.

### 4.1. Co-Location

Co-location consists of identifying the reflectance values (i.e., the pixels of the input image in each of the input bands) that correspond to each physical estimate. This is a non-trivial operation as physical

estimates are provided along irregularly shaped wave traces with varying patch sizes of >500 m (as shown in Figure 2), while reflectance is computed over much smaller patch sizes of 80 m. Our method assumes that a physical estimate is the most characteristic at the centre of its patch and, accordingly, computes the mean reflectance over an 80-m patch placed in the middle of each larger physical patch, separately for each of the bands, as illustrated in Figure 4.

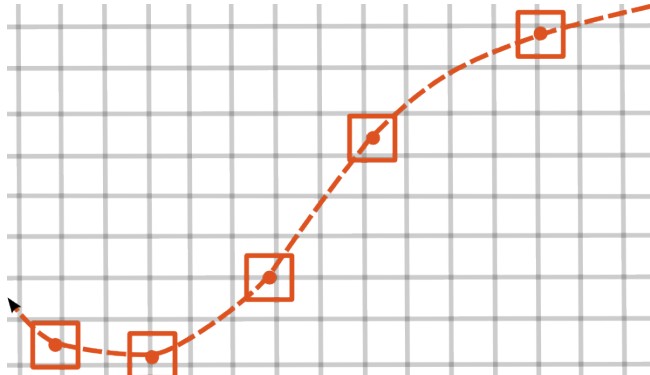

**Figure 4.** Illustration of the co-location method: the dashed orange curve represents a wave trace, on which each orange square covers the centre of a patch computed by the wave tracing method. From each band a mean reflectance value is computed for each of these squares to be used for training. The size of these squares is equal to the final resolution, as shown by the grid in grey.

The result consists of as many reflectance values as there are input bands (four in our case), co-located with each physical bathymetry estimate. As the water colour inversion method is not applicable beyond 12 m of depth [4], we co-locate only the physical estimates that are shallower than this threshold.

## 4.2. Candidate Training Set Generation

Our experiments have shown that training with the maximum amount of data available did not necessarily guarantee the best results, and that the statistical estimates could be sensitive to the way training values were selected. We therefore need a way to determine (1) how many co-located points to use for training; and (2) how to select these points.

Setting the size of the training set to a constant value is not an ideal solution as the optimal size depends on a number of variables, such as:

- the uncertainty of training data (0–15% of the ground truth for physical estimations between 0 and 20 m);
- the presence of clouds which can compromise the training;
- the amount of estimates provided by the physical model in the shallow water zone, which provides an upper bound on the number of training points that can be generated.

Rather than fixing the training set size *a priori*, we generate several *candidate training sets* of different sizes. We start with a size of 50 points that we increment by 50 up to about 1000, which is close to the maximum amount of training points we are typically able to generate for a single site using the physical model.

An entirely random sampling of training points may result in a significantly non-uniform depth distribution in the input, which in turn could generate a biased model. Our solution is rather to train models with reasonably uniform inputs that are achieved by randomly selecting the same amount of training points for ten depth intervals between 3 m and 12 m (the zone of applicability of our reflectance-based method).

*4.3. Training of Candidate Models*

For each candidate training set, we train a candidate machine learning model using *Gaussian Process Regression*, a Bayesian learning tool. Ref. [14] defines Gaussian processes as *a collection of random variables, any finite number of which have a joint Gaussian distribution.* The statistical properties deduced from this process define a model that is able to provide an estimate as well as its variance with respect to new inputs [19]. The output of variance was the main reason for our choice of GPR over other machine learning methods, on which we rely for the optimisation of the trained model.

As in many other machine learning techniques, GPR uses kernel methods to characterise nonlinear relationships between inputs and output [16]. This implies defining a kernel function describing the covariance function over pairs of data. Our chosen covariance function is the squared exponential, commonly used in the literature for similar applications.

Formally, the output target $\widehat{y}$ and its variance $\sigma^2$ are predicted by minimising the hyper-parameters of the considered kernel function using the training data (inputs and targets). The training inputs are symbolised by $\mathbf{x_n}$ and the training targets by $\mathbf{y_n}$. The dimension of $\mathbf{x_n}$ depends on the number of features considered. In this case, the water depth is the output target $\widehat{y}$, the training targets $\mathbf{y_n}$ are the estimation from the physical model, and the training inputs $\mathbf{x_n}$ are the mean reflectance of each band co-located with the training targets. The computational complexity of the minimisation operation increases with the amount of training data. For space reasons, we describe only briefly the argument needed for applying the approach with a new sample $\mathbf{x_{n*}}$. For a detailed explanation of GPs, we advise the reading of [14].

$$\widehat{y} = k_*^T \cdot (\mathbf{x_n}, \mathbf{x_n}) + \sigma_n^2 I]^{-1} \cdot y_n \tag{2}$$

$$\sigma^2 = \sigma_n^2 + k(\mathbf{x}_{n*}, \mathbf{x}_{n*}) - k_*^T \cdot [K(\mathbf{x_n}, \mathbf{x_n}) + \sigma_n^2 I]^{-1} \cdot k_* \tag{3}$$

where $K(\mathbf{x_n}, \mathbf{x_n})$ is the covariance matrix of the training sample $\mathbf{x_n}$, $k_*$ are the covariance values between the training sample $\mathbf{x_n}$ and the new sample $\mathbf{x}_{n*}$, $k(\mathbf{x}_{n*}, \mathbf{x}_{n*})$ are the covariance values of the new sample $\mathbf{x}_{n*}$, and $\sigma_n^2$ is the variance of the training data. $I$ represents the identity matrix.

*4.4. Selection of the Optimal Model*

In order to select the best model, we run an estimation on each candidate model trained in the previous step. Besides bathymetry estimates, GPR also outputs estimates of variance. In order to determine the optimal model we use the following criteria, in the order of decreasing priority:

1.　the number of high-variance estimates (the lower the better);
2.　the *spatial variability* of bathymetry estimates (the lower the better);
3.　the size of training data (the lower the better).

Firstly, high-variance estimates are those for which the variance output by the GPR exceeds a certain threshold (we have been using 2 m as threshold for standard deviation). The lower the number of such points, the more confident we are of the overall output. We thus generally prefer the model that produces the smallest number of high-variance estimates.

Secondly, the notion of *spatial variability* is motivated by the consideration that a 'smoother' output (in terms of adjacent bathymetry estimates) is more coherent with our training data than an output that shows rapid changes in water depth. While the GPR output is set up to produce results on a $80 \times 80$ m grid, the physical estimates that were used for training only have a resolution of 500 m at best. We thus consider that a very high spatial variability in the output is a warning sign on the reliability of the model. We define spatial variability $z$ as the mean of the absolute differences between an estimated value $\widehat{y}$ and its eight neighbouring values $y_n$ in the grid.

$$z = \frac{1}{8} \sum_{n=1}^{8} |\hat{y} - y_n| \qquad (4)$$

Thirdly, in case of similar results in terms of variance and spatial variability, we choose to minimise the size of data used for training. This is to minimise the time necessary for subsequent estimations using the same GPR model, as in the case of applying it to related images (see Section 7).

## 5. Statistical Bathymetry Estimation

The last step of the overall process in Figure 1 is the estimation step that uses the statistical model trained and selected as the best in the previous step. The input of the model is a multispectral image with the same bands as those used for training.

While the primary goal of the statistical model is to produce high-resolution and high-coverage estimates for the very same image on which it was generated, we also present a different but very important use case of applying the model to other acquisitions. In particular, the statistical model is not bound by the input constraints of the physical model and can therefore be applied to images without any exploitable swell. This allows us to extend the applicability of our method considerably.

However, applying the trained model to different images implies that the physical conditions latent in our reflectance data used for training—such as atmospheric conditions, seabed properties, and water quality—stay similar to a large extent across the images. This is a strong constraint that limits the scope of application of the trained model to what we called *related images* in Figure 1: images of the same zone taken at different times, or images of nearby zones taken in a short time interval. An example of the former case can be the monitoring of changing bathymetry (e.g., moving seabeds) of a given site even in the absence of swell. The latter case corresponds to reusing the trained model in nearby sheltered areas that are rarely or never exposed to the swell.

## 6. Experimental Validation

In this section we evaluate each of the major steps of our method: the physical model, the optimisation of the statistical model, and finally the output of the statistical model as the result of the entire setup.

### 6.1. Evaluation Sites and Data

The geographical region for our experiments was chosen to be Hawaii due to the availability of ground truth data for several sites. Furthermore, Hawaii benefits both of a long-period swell (more than 8 s), regularly, necessary for the physical model, and of clear waters that are beneficial for the statistical model.

Three sites have been used along the North-East coast of the island of Oahu: *Waimanalo*, *Kaneohe*, and *Kailu* (see Figure 5). Each cover an area of about 8 × 8 km in a nearly contiguous way along a 30 km-long portion of the coast.

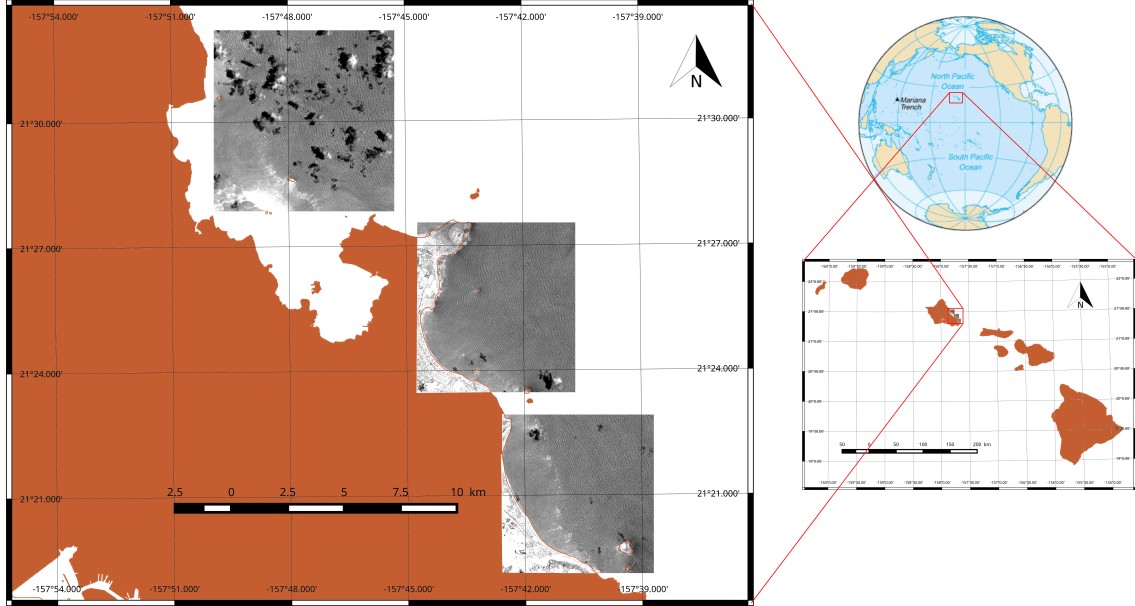

**Figure 5.** Location of the three sites from North to South: *Kaneohe*, *Kailu*, and *Waimanalo*, along the North-East coast of the island of Oahu. The satellite image patches are extracted from the Sentinel-2 image acquired 24 May 2016.

The ground truth bathymetry was measured in 2000 by a SHOALS airborne LIDAR campaign (http://www.soest.hawaii.edu/coasts/data/oahu/shoals.html) [20]. Its horizontal accuracy is ±3 m and its vertical accuracy is about ±0.15 m. Because of the volcanic nature of the Hawaiian Islands, the time interval between the LIDAR campaign and the satellite acquisitions is not considered as a bias factor.

Four Sentinel-2 images of size 100 × 100 km were selected for the evaluation, each covering all three sites. As it can be seen from Table 1, of the 12 site-specific acquisitions we only used nine, as the remaining three acquired on 2 and 12 August 2016 suffered from a cloud coverage too dense to be exploited.

### 6.2. Results of the Physical Model

Table 1 presents for each site the error measures with respect to ground truth. Error measures are presented in two forms between 0 and 20 m of depth: (1) along wave rays; and (2) interpolated over the entire site. We interpolated the results of the physical model to a regular grid of 80 m in order to be able later to perform meaningful comparisons between the physical and the GPR results.

The table also shows the number of physical estimations realised along the ray traces. These values are significant given that a minimal number of estimates (more than 50) is required to train a GPR model efficiently. In the case of 3 July at Waimanalo and 2 August at Kaneohe, the number of training points is insufficient to train a GPR model efficiently. For these two days, the presence of clouds explains why the method is not working properly: the clouds interrupt the wave traces and, consequently, the approach breaks down. This explains the low coefficients of correlation.

Finally, Table 1 also includes an estimation of the tidal heights at acquisition time. These values were provided by the *SHOM* (*Service Hydrographique et Océanographique de la Marine*. Their online tide forecast can be found at http://maree.shom.fr/) [21]. Depending on the tide, the bathymetry obtained may present a bias from one acquisition to another. In the case of Hawaii, the tidal shift reaches 75 cm. In other regions, however, it may be superior to 5 m, such as in the Bay of Mont Saint-Michel in France. In our evaluations we did not take tidal heights into account.

**Table 1.** Comparisons between physical model results and ground truth for four Sentinel-2 acquisitions over three sites. Comparisons are provided along wave rays as well as over an interpolated grid. Tidal heights are indicated at the acquisition time at Honolulu harbour.

| Date | Tidal Height | Site | Comparisons along Wave Rays | | | | After Interpolation | | |
|------|------|------|------|------|------|------|------|------|------|
| | | | $R^2$ | R | RMSE | # Estimates | $R^2$ | R | RMSE |
| 24 May 2016 21:19 UTC | −0.01 m | Waimanalo | 0.87 | 0.93 | 3.47 | 1772 | 0.85 | 0.92 | 2.07 |
| | | Kaneohe | 0.77 | 0.88 | 3.42 | 1226 | 0.52 | 0.72 | 3.74 |
| | | Kailua | 0.75 | 0.87 | 4.93 | 1319 | 0.72 | 0.85 | 2.44 |
| 3 July 2016 21:19 UTC | 0.17 m | Waimanalo | 0.76 | 0.87 | 5.06 | 148 (clouds) | 0.64 | 0.80 | 3.66 |
| | | Kaneohe | 0.72 | 0.85 | 6.40 | 2014 | 0.71 | 0.84 | 2.77 |
| | | Kailua | 0.49 | 0.70 | 6.17 | 701 (clouds) | 0.61 | 0.78 | 2.83 |
| 2 August 2016 21:19 UTC | 0.14 m | Waimanalo | 0.45 | 0.67 | 6.14 | 593 (clouds) | 0.29 | 0.54 | 5.20 |
| | | Kaneohe | 0.38 | 0.62 | 5.28 | 171 (clouds) | 0.28 | 0.53 | 4.82 |
| | | Kailua | | | | n/a due to cloud coverage | | | |
| 12 August 2016 21:19 UTC | 0.52 m | Waimanalo | | | | n/a due to cloud coverage | | | |
| | | Kaneohe | 0.67 | 0.82 | 6.45 | 2114 | 0.59 | 0.77 | 2.82 |
| | | Kailua | | | | n/a due to cloud coverage | | | |

## *6.3. Statistical Model Optimisation*

As explained in Section 4.2, statistical estimates can be improved by computing the optimal training set size for each acquisition. In our evaluations, for each of our seven acquisitions (two out of nine having been excluded for an insufficient number of training points) we trained 18 candidate models of sizes of 50, 100, 150, ..., 900 points. For each trained candidate model, we computed the spatial variability of the estimation results (as explained in Section 4.2), as well as the correlation coefficient with respect to the ground truth in order to validate the optimisation approach.

We obtained the following results:

- the number of estimations with a standard deviation under 2 m is not increasing with the number of training points;
- beyond 150 training points, the correlation coefficients with respect to the ground truth do not change significantly (around $0.87 \pm 3\%$ in our evaluations, see Figure 6);
- estimations show an increasing spatial variability with the number of training points (see Figure 7a,b).

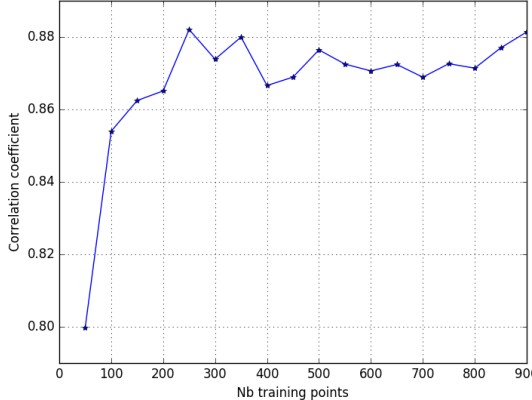

**Figure 6.** Correlation coefficient observed on one experiment in function of the number of training points.

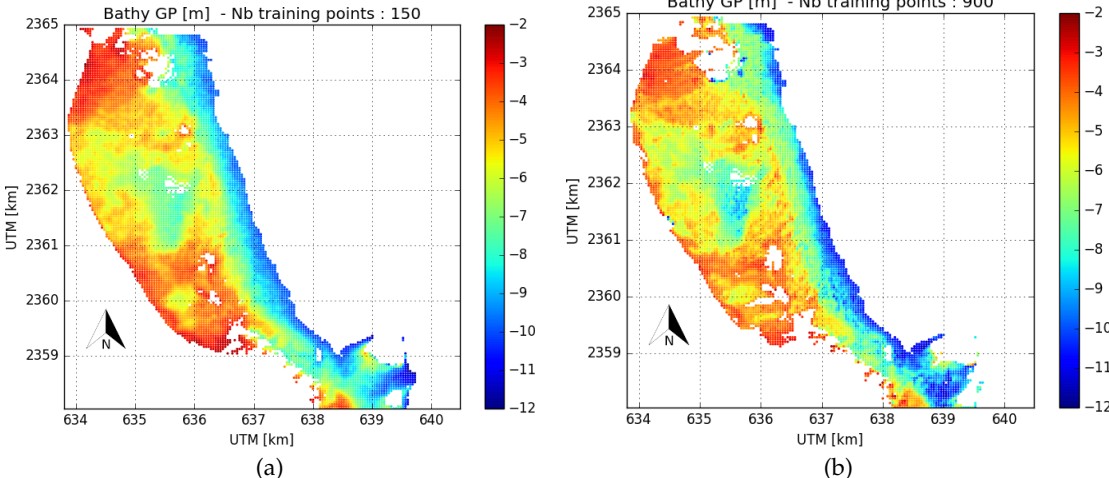

**Figure 7.** Examples of statistical estimates with (**a**) 150 and (**b**) 900 training points. Spatial variability is visibly higher on the 900-point model.

Overall it turned out that all models between 150 and 900 training points present robust and roughly equivalent solutions in terms of correlation coefficients and RMSE. Following the principles put forth in Section 4.2, we opted for smaller models in the range of 150–300 points, depending on the site, for lower spatial variability and faster computing times.

### 6.4. Results of the Statistical Model

As it can be seen on Table 2, the GPR model results have a signficiantly and systematically higher correlation coefficient and a smaller root mean square error to ground truth compared to the physical model (interpolated over the same $80 \times 80$ m grid). We have to precise here that the interpolated results from the physical model are slightly different from Table 1 since only the estimation between 2 and 12 m depth are considered.

Furthermore, the GPR model provides a significantly higher resolution and a nearly complete coverage of the shallow water zone, fulfilling our main objectives. These results are visualised in Figure 8.

**Table 2.** Comparisons of GPR and physical model results to ground truth, between 2 and 12 m of depth.

| | | GPR | | | Physical Model | | |
|---|---|---|---|---|---|---|---|
| **Date** | **Site** | **$R^2$** | R | **RMSE** | **$R^2$** | **R** | **RMSE** |
| 24 May 2016 | Waimanalo | 0.79 | 0.89 | 1.31 | 0.49 | 0.70 | 1.94 |
| | Kaneohe | 0.61 | 0.78 | 2.47 | 0.46 | 0.68 | 3.69 |
| | Kailua | 0.76 | 0.87 | 1.42 | 0.56 | 0.75 | 2.05 |
| 3 July 2016 | Waimanalo | 0.49 | 0.70 | 2.42 | 0.38 | 0.62 | 3.80 |
| | Kaneohe | 0.69 | 0.83 | 1.68 | 0.59 | 0.77 | 1.98 |
| | Kailua | 0.64 | 0.80 | 1.82 | 0.52 | 0.72 | 2.24 |
| 2 August 2016 | Waimanalo | 0.28 | 0.53 | 3.16 | 0.11 | 0.33 | 5.38 |
| | Kaneohe | 0.23 | 0.48 | 3.10 | 0.21 | 0.46 | 5.28 |
| | Kailua | no results from physical model | | | | | |
| 12 August 2016 | Waimanalo | no results from physical model | | | | | |
| | Kaneohe | 0.76 | 0.87 | 1.6 | 0.42 | 0.65 | 2.18 |
| | Kailua | no results from physical model | | | | | |

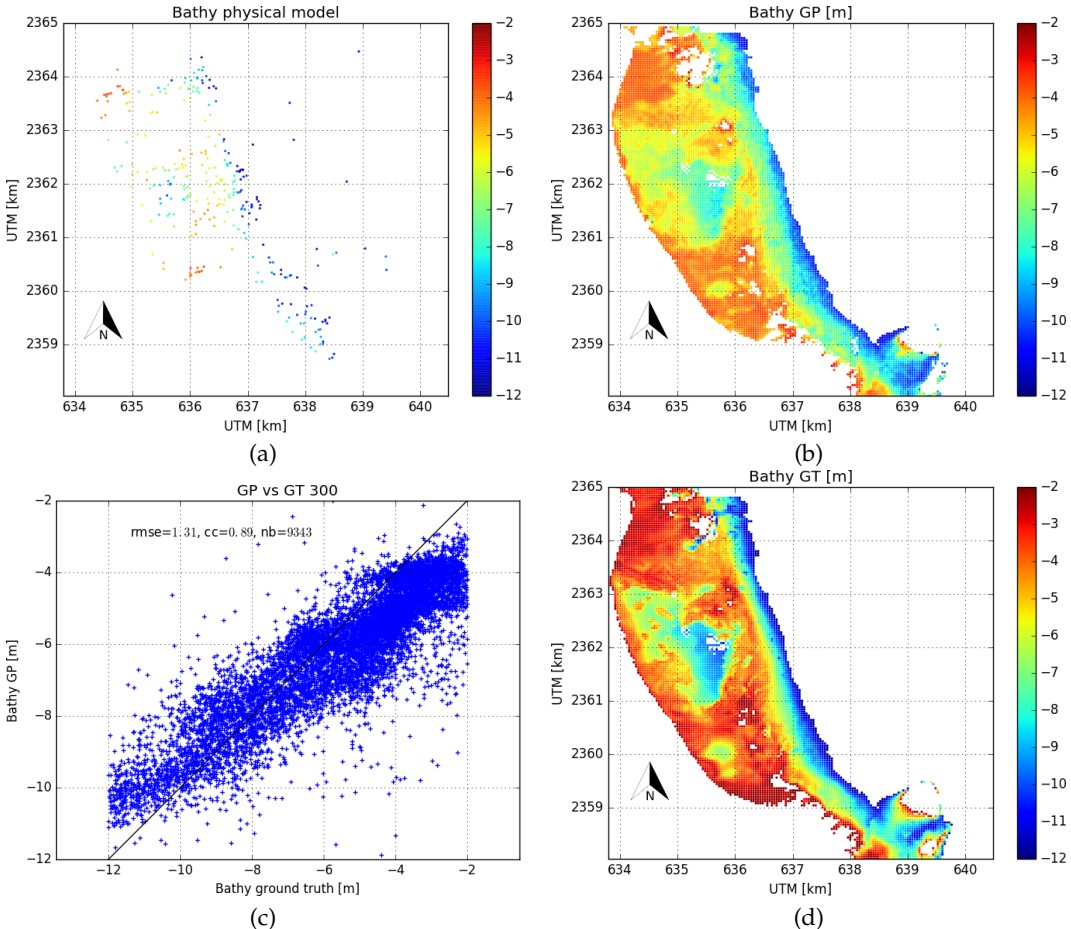

**Figure 8.** Example of Waimanalo beach showing the improvement in bathymetry estimation using the GPR model: (**a**) bathymetry obtained by the physical model (300 values between 2 and 12 m); (**b**) GPR results; (**c**) scatterplot between the GPR results and the ground truth; and (**d**) geographical representation of the ground truth.

These first results clearly show how a statistical approach can significantly improve on the estimates provided by the physical model. These results, however, largely depend on the accuracy of the latter. In our experience, as long as the physical model itself proposes coastal estimates that are accurate enough (correlation coefficient with the ground at least 0.7), the precision of the GPR results will remain largely acceptable, as shown by graph (c). In the contrary case, the statistical model is less likely to produce a reliable solution, as it can be seen for the acquisition of 2 August.

## 7. Spatial and Temporal Transfer of the Statistical Model

The results obtained in the previous section underline the fact that the accuracy of the statistical model depends on that of the physical model it was trained on. In this section we evaluate and discuss the possibility of applying a high-quality trained model to other acquisitions, offering a way to avoid having to use the physical model where physical conditions are less favourable (e.g., due to a presence of clouds or an absence of swell).

Two kinds of experiments were conducted:

- *spatial transfer* consists of applying the statistical model to a site nearby yet different to where it was trained, on the same day of acquisition;
- *temporal transfer* consists of applying the statistical model to acquisitions on different dates, on the same site.

The goal of the experiments was to explore the robustness of the trained model with respect to physical conditions changing in space and in time.

### 7.1. Spatial Transfer

We expect a GPR model to be spatially transposable if atmospheric conditions, seabed properties, and water quality are very similar between the site where it was trained and where it is applied. As the three sites of our evaluations (Waimanalo, Kailu, and Kaneohe) are close to each other (less than 20 km of distance see Figure 5), these conditions have a good chance to be met on one acquisition. So we are expecting good results for spatial transfer from one site to another on a same day.

For our experiment we chose three of the four acquisitions, the one of 2 August not having been considered because of its poor results (presence of clouds). For each acquisition we chose the best statistical model from the three possible sites, based on the results from Table 2. The model was then applied to the two other sites for bathymetry estimation.

From Table 3, it can be seen that the results remain equivalent or even better with respect to the non-transferred case (see Table 2 for comparison). Even more interestingly, transposing the model trained on the 12 August acquisition at Kaneohe to the site of Kailua, we managed to obtain good-quality results (correlation coefficient 0.78, RMSE 2.14 m), which had been impossible using the physical model on the same site due to the configuration of clouds.

Our results clearly point in the direction that a statistical model can be successfully transposed across sites that present similar conditions with respect to the atmosphere, seabed, and water quality.

**Table 3.** Spatial transfer: comparisons between ground truth and GPR results obtained with a model trained on the same acquisition day but on a different site.

| Date | Site | Best Model | $R^2$ | R | RMSE [m] |
|---|---|---|---|---|---|
| 24 May 2016 | Kaneohe | Waimanalo | 0.74 | 0.86 | 1.48 |
| | Kailua | Waimanalo | 0.79 | 0.89 | 1.63 |
| 3 July 2016 | Waimanalo | Kaneohe | 0.48 | 0.69 | 2.20 |
| | Kailua | Kaneohe | 0.58 | 0.76 | 1.91 |
| 12 August 2016 | Kailua | Kaneohe | 0.61 | 0.78 | 2.14 |

### 7.2. Temporal Transfer

Reusing a statistical model on the same site but over several acquisitions across time could potentially provide an efficient solution for coastal monitoring, addressing the issue of the non-applicability of the physical model due to no swell or the presence of clouds. As in the case of spatial transfer, the underlying assumption is that the physical conditions at the times of acquisition remain similar. While we expect seabed properties to be largely (even if not completely) invariant on the same site, this may not be the case with respect to atmospheric conditions or water quality.

Among the 17 test cases presented in Table 4, only three were able to improve the results with respect to models trained on the same acquisition. Most of the results are weak when compared to ground truth. These results hint at the conclusion that, in the absence of evidence for the invariance of physical conditions, the reusability of the statistical model is generally limited to a very short time interval (acquisitions on the same day).

However, in certain cases the condition of temporal locality could be relaxed: when an image is very difficult or impossible to process through the physical model, and a less-than-optimal result is acceptable. As it can be seen from Tables 1 and 2, this is the case of the acquisitions of 12 August at Kailua (no result from the physical model) and of 2 August at Waimanalo and Kaneohe (correlation coefficients 0.48–0.53). In these cases, models temporally transposed from 3 July provide correlation coefficients of 0.63–0.68 and RMSE of 1.85–2.10 m. The accuracy remains relatively weak yet significantly improved with respect to those obtained through the physical model.

**Table 4.** Temporal transfer: comparisons between ground truth and GPR results obtained with a model trained on the same site but on different days of acquisition.

| Date | Site | Model Trained on | $R^2$ | R | RMSE [m] |
|---|---|---|---|---|---|
| 24 May 2016 | Waimanalo | 3 July | | no result | |
| | Kaneohe | 3 July | 0.40 | 0.63 | 3.09 |
| | Kaneohe | 12 August | 0.32 | 0.57 | 2.46 |
| | Kailua | 3 July | 0.12 | 0.34 | 3.16 |
| 3 July 2016 | Waimanalo | 24 May | 0.21 | 0.46 | 2.61 |
| | Kaneohe | 24 May | 0.45 | 0.67 | 4.19 |
| | Kaneohe | 12 August | 0.40 | 0.63 | 2.59 |
| | Kailua | 24 May | 0.01 | 0.08 | 1.67 |
| 2 August 2016 | Waimanalo | 24 May | 0.12 | 0.34 | 2.38 |
| | Waimanalo | 3 July | 0.42 | 0.65 | 1.85 |
| | Kaneohe | 24 May | 0.19 | 0.44 | 4.05 |
| | Kaneohe | 3 July | 0.46 | 0.68 | 2.10 |
| | Kaneohe | 12 August | 0.20 | 0.45 | 2.84 |
| 12 August 2016 | Kaneohe | 3 July | 0.55 | 0.74 | 2.14 |
| | Kaneohe | 24 May | 0.19 | 0.44 | 4.79 |
| | Kailua | 24 May | 0.01 | 0.09 | 3.93 |
| | Kailua | 3 July | 0.40 | 0.63 | 1.99 |

## 8. Conclusions

In this article, we demonstrated how the combination of physical and statistical methods applied to satellite acquisitions can considerably improve coastal bathymetry estimates, without relying on any other information than the image itself. Without the physical model, the statistical model would not have training values and without the statistical model, the resolution of estimates remains weak which limits practical exploitability.

Our approach estimates coastal bathymetry on a regular grid of 80 m and for depths between 2 and 12 m. The overall performance strongly depends on the results of the physical model. When the latter is not perturbed by an absence of swell or a too important presence of clouds, the statistical model clearly improves the resolution and the coverage of the result.

In principle, the reuse of trained statistical models for new sites or for the same sites over time has enormous potential for coastal monitoring. However, the statistical models trained by our method encode atmospheric and oceanic conditions that are specific to the training acquisition. This fact theoretically limits the reusability of trained models to acquisitions that are either spatially or temporally close. We have demonstrated that predictions obtained from a statistical model trained on neighbouring areas are able to provide similar or even better results than the ones obtained on the same location. Temporal transfers, on the other hand, provided generally weaker results that we consider usable only to provide depth approximations in the absence of any other estimation method.

Our experiments used multispectral Sentinel-2 images exclusively. As we mentioned earlier, other instruments such as Landsat-8 have similar multispectral capabilities. It is therefore possible

to use our method with multiple sensors simultaneously, in the goal of increasing the temporal coverage for coastal monitoring.

**Author Contributions:** Conceptualization, C.D.; Data curation, C.D.; Investigation, C.D.; Methodology, C.D. and F.M.; Software, C.D.; Supervision, F. M.; Validation, C.D.; Visualization, C.D.; Writing—original draft, C.D.; Writing, review and editing, C.D.

**Funding:** This research has been founded by the European Space Agency for the ECOBAW project in the framework of the Living Planet Fellowship.

**Acknowledgments:** This research has been founded by the European Space Agency for the ECOBAW project in the framework of the Living Planet Fellowship. We would like to thank the University of Dortmund that provided the *pyGPs* tool, the U.S. Geological Survey for providing Landsat-8 images, the European Space Agency for the availability of Sentinel-2 acquisitions, the University of Hawaii and NOAA for SHOALS data, and the French Service of Hydrography and Oceanography of the Marine for providing tidal heights.

**Conflicts of Interest:** The authors declare no conflict of interest.

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
