# Peer review of "High-Coverage Satellite-Based Coastal Bathymetry through a Fusion of Physical and Learning Methods"

_remotesensing, doi:10.3390/rs11040376_

Round 1
Reviewer 1 Report
Danilo & Melgani, 2018
High-Coverage Satellite-Based Coastal Bathymetry through a Fusion of Physical and Learning Methods
This paper is breaking new ground for satellite derived Bathymetry (SDB) with a radically new approach for calibrating the radiance-to-depth inversion.
The experimental results are impressive and well presented. I recommend this paper for publication with minor revisions suggested below.
Abstract: replace ‘several sites around the Hawaiian Islands‘ to ‘3 sites around the island of Oahu, Hawaii.‘
Page 2: References [1-6] required in situ calibration. All these are pre 2005. More recent works claim to have eliminated the need for in situ calibration. See for example
https://www.eomap.com/exchange/pdf/EOMAP_Satellite_Derived_Bathymetry_Whitepaper.pdf
Page 3. the paper states ‘We did not apply any preprocessing to the images provided by the instrument.’ What does prepreprocessing mean in the context of this work? Normally SDB starts with atmospheric corrected bottom-of-atmosphere calibrated radiometry. It appears that the fusion method eliminates the need for atmosphere correction – which is a very nice feature in itself.
Page 3. The meaning of ‘regular swell’ is not clear. This is not a standard term used to describe waves.
Page 4. The paper states: ‘The model provides bathymetry estimates within the range of about 3–25m of depth: beyond these boundaries the linearity of the physical model does not hold anymore.’ The linear model is valid for h<<depth, where h is the wave height. So one could say the linearity breaks down for depth <3m. But linearity still holds for depths>25 m.
Page 5. The paper states: ‘the water colour inversion method is not applicable beyond 12 m of depth [4].’ The 12 m limit in [4] may be due to the fact that they used SPOT images which has 8 bits. Sentinel 2 has 12-bit radiometry and there are claims of reaching 20-30 m.
Page 6. Does GPR training respond to changing substrate types? For example sand, grass, reef. In conventional SDB the different substrates are separated. Does GPR separate substrates in a fully automated way? If true, this is an additional benefit of GPR.
Pages 6-7. It would be easier to follow the explanation of GPs if the terms xn, yn, and y-hat are related to the specific inputs. E.g., the radiance levels of each band, the depth estimated from waves, and the depth derived after fusion.
In equations (2) and (3): should the ‘.’ be raised?
Page 10, Figure 6. Very impressive results!
Tables 2, 3, 4, and 5. The tables include both R and R^2. Do you need to tabulate both?
Tables 3, 4, and 5: include the year in the dates (if possible use the same date format as in Table 2)
Page 12-13. Why did temporal transfer not work out well? One possible explanation is that the image data was not corrected for atmosphere. Another explanation might be that there are variations in water attenuation from one image to the next. Do the authors have some idea about this?
Author Response
pre.cjk { font-family: "Courier New", monospace; }p { margin-bottom: 0.25cm; line-height: 120%; }

Reviewer 2 Report
Revision of the manuscript n. 427382
I have read the manuscript entitled “High-Coverage Satellite-Based Coastal Bathymetry through a Fusion of Physical and Learning Methods” by Danilo and Melgani. The paper provides a satellite-based coastal bathymetry obtained through a matching of Physical and Learning Methods, specifically using the combination of a linear wave model and a statistical model with Gaussian Process Regression (GPR).
The paper is well organized, and the aims are well exposed in the Introduction section. I only suggest improving the citation of other references on the argument. Method overview section comprising the three main steps is clear. However, I find that sections 3 (Physical Bathymetry Estimation), 4 (Statistical Model Optimisation and sub-sections therein), and 5 (Statistical Bathymetry Estimation) contain a discussion on the applied methodology and then my suggestion is to join them as a sub-section of the Section 2 (Method overview). Thus, as an example the Section 3 (Physical Bathymetry Estimation) will be numbered as sub-section 2.1, and so on. This choice will allow the reader to follow your methodology in a single section, on the whole, and to analyse the results in the next section. In fact, section 6 (Experimental validation) could be re-named for example Results of the Experimental validation, or differently. In this case, you must modify the heading of both the sub-section 6.2 and 6.4. Moreover, I think it is useful to add a new figure in sub-section 6.1 (Evaluation Sites and Data) showing the location of your study area that lack in the paper. Discussion section (Spatial and Temporal Transfer of the Statistical Model) provides mainly results of two experiment but it is a bit stressed and there is not critical argumentation on the experiment conducted in section 7 (Spatial and Temporal Transfer of the Statistical Model). Moreover, there is a lack of comparison with scientific literature on the argument. I think this latter section must be more detailed rewritten for an international audience because it is not enough for the purpose of the Journal. Conclusions are sufficient and coherent with the aims of the paper. Considering the whole text, I think that is necessary a significant improvement because of many sentences that are unclear and a bit stressed. Figures and tables are quite sufficient, but it is important to add a figure showing the location of the study area. Finally, I suggest a deep revision of the text by a native English speaker. In conclusion, the manuscript is suitable for publication on the Remote Sensing Journal, after major revisions. Please see the below detailed comments.
Detailed comments
Line 99: please, delete “below” and add “in Fig. 1”. Add after “step 3” the word “in Fig. 1”.
Line 102: delete “In [11] the authors have” and add “Danilo et al. (2006) [11]”.
Line 108: delete “in” and add “by”.
Lines 116-120: the phrasing is not clear, please check the English.
Line 122: delete “to our earlier paper [11]” and add “to paper by [11]”.
Line 162: delete “—that we present in section 6.3—”.
Lines 164-165: please, rephrasing.
Lines 174-176: please, rephrasing in a correct English.
Lines 240-241: please delete.
Lines 247-248: please, rephrasing.
Line 252: It is useful to add a figure showing the location of sites you cited.
Lines 270-271: where you take this information? Please cite the references or justify it.
Line 278: “…For our evaluations, for each of our seven acquisitions…” is not a good English. Please, rewrite it.
Line 292: check.
Lines 306-307: please rewrite it.
Line 311: please, check the grammar.
Lines 322-323: please I don’t know, in the spatial transfer sub-section you write that image acquisition and evaluation is valid if the day of acquisition is the same for all images. How do you explain the different day of acquisition of images reported in Table 4?
Author Response
See the pdf document

Reviewer 3 Report
This is a very interesting paper that the authors have combined the physical linear wave model and the statistical model to predict the bathymetry from the satellite imagery. Different from the conventional methods requiring the in-situ data to establish the relationship between the spectral observation with the physical water depth, the use the physical model has provided estimates that are readily used for the statistical model. Even though the physical model doesn’t generate high accuracy of estimates, through the criteria set by the authors, good handling of the estimates can compensate the shortcoming of the physical model and generate good enough estimates for some applications. I really enjoyed reading this manuscript and I do believe it deserves the publication.
Some minor comments:
1. Line 131, provided the original spatial resolution for Sentinel-2 is 10m, how do you determine the optimal output resolution is 80m for the bathymetry for the purpose of suppressing the wave reflectance and noise?
2. Line 285, why do you think the increasing spatial variation is bad for the bathymetry estimation?
Author Response
See the reply to the review in the pdf file

Reviewer 4 Report
This work presents an update regarding the work already published by the same authors. The work is clearly presented and the methods are robust and well justified. My major doubt is related to the term “unsupervised”. It is really an unsupervised method? If so, why you used a training data set?
Some sentences need references in order to support the affirmations.
Section 2 should give more information on the methods used.
See my minor comments in the pdf file.

Author Response

(The authors gave the same response as above.)

Round 2
Reviewer 2 Report
The paper is suitable for publication in the Journal.